# Using Soil Water Status Sensors to Optimize Water and Nutrient Use in Melon under Semi-Arid Conditions

**Susana Zapata-García** [1] , **Abdelmalek Temnani** [1] , **Pablo Berríos** [1] , **Pedro J. Espinosa** [2], **Claudia Monllor** [3] and **Alejandro Pérez-Pastor** [1,*]

1   Departamento de Ingeniería Agronómica, Universidad Politécnica de Cartagena (UPCT), Paseo Alfonso XIII, 48, ETSIA, 30203 Cartagena, Spain; susana.zapata@upct.es (S.Z.-G.); abdelmalek.temnani@edu.upct.es (A.T.); pablo.berrios@edu.upct.es (P.B.)
2   Europe, Middle East & Africa Region (EMEA) Plant Health Portfolio, FMC Agricultural Solutions, 28046 Madrid, Spain; pedro.espinosa@fmc.com
3   Plant Health Portfolio, FMC Agricultural Solutions, 28046 Madrid, Spain; claudia.monllor@fmc.com
*   Correspondence: alex.perez-pastor@upct.es

**Abstract:** Nowadays, agriculture must satisfy the growing demand for food, and increasing its sustainability, from an environmental, economic, and social point of view, is the only way to achieve this. The objective of this study was to increase the water and nutrient use efficiency of a melon crop during two consecutive seasons under commercial conditions, growing under semi-arid area. For this purpose, two treatments were studied: (i) a farmer treatment (FRM), fertigated at ~100% of crop evapotranspiration ($ET_c$) during the whole growing season; and (ii) a precision irrigation treatment (PI), irrigated by adjusting, between flowering and ripening, the weekly farmer irrigation to minimize the leaching below the root system. The threshold for allowable soil water depletion in the active root uptake zone was set at 20–30%. The cumulative water savings in each year relative to the FRM treatment ranged between 30 and 27% for 2020 and 2021, respectively. Yield was not negatively affected, with no differences in fruit load (fruit per m) or fruit weight (kg) between irrigation treatments, although higher yields were obtained in the second year due to seasonal changes. The crop water status indicators evaluated (stem water potential, net photosynthesis, and stomatal conductance) were not affected by the irrigation treatments. Water and nitrogen productivity, on average, increased by 45.5 and 54.4% during the experimental period, respectively; the average PI ascorbic acid content increased by 33.4%.

**Keywords:** water scarcity; sensorization; leaching; sustainability; nitrogen; vitamin C

## 1. Introduction

Water is a vital resource for human life and ecosystems biodiversity but is also indispensable for sustainability of economic activities and development [1]. However, there is a high pressure on the use of water resources, which will be exacerbated by the climate change. The irrigated agricultural sector is the most demanding of water, accounting for about 70% of global water withdrawals [2] and is expected to face a complex challenge. Due to the rising temperatures, semi-arid Mediterranean areas such as the southeast of Spain will be the most affected, as they will have to extend the irrigation period of the crops, at the same time that plants' evapotranspiration will increase [3–6].

Along with water scarcity, we must face an increase in world population, with a growing demand for food. That fact has come to the attention of the United Nations, creating different Sustainable Development Goals [7] related to agriculture and rational water use. This underscores the need for recognizing the value of water in agriculture and making rational and sustainable use of it, through different measures to optimize its use and management, leading to an increase in the sustainability of irrigated agriculture [8,9].

The melon crop (*Cucumis melo* L.) would be considered as particularly sensitive to drought and water stress, as it would negatively affect vegetative growth and reduce the yield via fruit cracking [10], therefore assuming large economic losses. Spain is the main producer and exporter in the European Union, generating 63% of the European production, the amount of which totals a value of EUR 325 million [11].

Although the Region of Murcia has a semi-arid Mediterranean climate [12], it is the main Spanish producing area for plain skin melons, a category that includes 'Piel de Sapo' and yellow melons, which accounted for 68% of the national production during 2021 [13]. These areas, which have had a steady diminishing supply of water, have been able to maintain strong agricultural activity, which not only supplies the food necessary to feed a growing population, but also plays an important role in mitigating climate change through the carbon sequestration in the soil [14]. These facts evidence the need to increase agriculture sustainability to ensure the sector's activity, and improving the water use efficiency will minimize the environmental impact and increase the productivity of agricultural systems to achieve economic, social, and environmental benefits. Regarding this need, some authors such as Fabeiro et al. have analysed the sensitivity of the different phenological stages of melon to moderate or severe water deficit through controlled deficit irrigation techniques [15] or by identifying drought resistant traits to improve breeding strategies [16]. In addition to water, the nutrient use efficiency in melon has been studied through fertilization at different doses, as shown in Castellanos et al. [17].

As in any sector, digitalization would help to increase process efficiency. Specifically, sensorization in agriculture can help in coping with water scarcity by providing real-time data on soil moisture levels, weather conditions, and crop water requirements. This information can be used to optimize irrigation practices, reduce water waste, avoiding water leaching into groundwater, and ensure that crops receive the appropriate amount of water and nutrients at the right time. This can lead to more efficient water use, increased crop yields, and improved water management in areas facing water scarcity as reported by other authors [18,19].

The aim of our research was to determine the effect on water and nutrient use efficiency of irrigation scheduling using multi-depth soil sensors in melon crops under semi-arid conditions.

## 2. Materials and Methods

### 2.1. Study Sites and Experimental Conditions

The trial was carried out on melon (*Cucumis melo* L.) grown in two commercial farms located in the Region of Murcia (SE Spain) during 2020 (Farm 1) and 2021 (Farm 2). Table 1 details the experimental conditions of each farm used during the experimental period.

The crop evapotranspiration ($ET_c$) was calculated according to the FAO method [20], and the crop coefficients ($K_c$) for the initial, mid-season, and end stages were 0.5, 0.85, and 0.6, respectively [21]. Reference evapotranspiration ($ET_0$), rainfall, and vapor pressure deficit (VPD) were obtained from the agroclimatic station 'Torre Pacheco TP-42' belonging to the "Murcia Agrometeorological Information Service" network (37°46′26″ N 0°53′55″ W) [22].

The nutritional requirement was determined on the basis of the technical recommendations for crop fertilization for the crop and according to the current legislation for nitrogen fertilization in the study area, Campo de Cartagena [23,24]. The most restrictive extraction coefficients for open field melon cultivation were considered: 3.5, 1.4, and 7.1 kg N, $P_2O_5$, and $K_2O$ per ton of harvested fruit, respectively [4].

Finally, a nutritional requirement for both farms was determined considering the soil and irrigation water nutrient inputs, being 87, 24, and 48 kg ha$^{-1}$ and 79, 24, and 75 kg ha$^{-1}$ for N, $P_2O_5$, and $K_2O$, for Farm 1 and 2, respectively.

A randomized experimental design was established for both farms with four repetitions, each one composed of three adjacent rows of eight plants. The central row was monitored as the experimental unit and the others served as a plant border. Two irrigation treatments with four replicates each were tested: (i) farmer (FRM), irrigated to satisfying

crop needs during the entire cycle, according to the $ET_c$; and (ii) a precision irrigation treatment (PI), irrigated by adjusting, between flowering and ripening, the weekly farmer irrigation to minimize the leaching below the root system. The threshold for the permissible soil water depletion in the zone of active root absorption (30 cm depth) was set between 20 and 30% of field capacity. The evolution of the soil water content (SWC) was measured with FDR-type sensors. As the fertigation was applied during the irrigation events, the reduction in the fertilizer doses was proportional to the reduction in irrigation water for each day. The amount of water applied in each treatment was controlled via volumetric water meters.

**Table 1.** Experimental conditions of each commercial farm used during the study years.

| | Farm 1 | Farm 2 |
|---|---|---|
| Year | 2020 | 2021 |
| Location | Torre Pacheco (Region of Murcia, SE Spain) 37°45′58″ N 0°58′03″ W | 37°47′18″ N 1°2′54″ W |
| Cultivar | Cordial F1 (Sakata Seeds) | Valderas F1 (Clause Vegetable Seeds) |
| Growing cycle | 90 days (30 April to 29 July) | 91 days (7 April to 7 July) |
| Cultivation system | Geotextile micro tunnel of polypropylene fibres for thermal protection (0.5 m high in the middle) and transparent plastic mulch. | |
| | 1.8 m × 1.6 m planting frame 3472 plants per ha | 1.8 m × 1.3 m planting frame 4273 plants per ha |
| Irrigation system | Drip irrigation system with one drip line per row and emitters spaced at 0.3 m with a flow rate of 2.2 L h$^{-1}$. | |
| Standard cultural practices | The fertilization, weed, pest, and disease control program was carried out according to commercial management using the usual criteria for the productive zone. | |
| Soil characteristics | The soil profile up to 0.3 m depth corresponded to a clay loam texture class (39% sand, 22% silt, and 39% clay), a bulk density of 1.40 g cm$^{-3}$, a 1.4% of organic matter, and an $EC_{1:5}$ of 0.613 mS cm$^{-1}$. The estimated field capacity and wilting point values based on Saxton et al. [25] were 36.1% and 23.8%, respectively. The [1] CEC was 11.2 meq 100 g$^{-1}$ and the soil nutrients concentrations were 1.07, 0.11, 0.59 g kg$^{-1}$ for N–$P_2O_5$–$K_2O$, respectively. | The soil profile up to 0.3 m depth corresponded to a silty clay texture class (14% sand, 44% silt, and 42% clay), a bulk density of 1.35 g cm$^{-3}$, a 1.7% of organic matter, and an $EC_{1:5}$ of 0.303 mS cm$^{-1}$. The estimated field capacity and wilting point values based on Saxton et al. [25] were 39.7% and 25.2%, respectively. The [1]CEC was 16.5 meq 100 g$^{-1}$ and the soil nutrients concentrations were 1.16, 0.09, 0.33 g kg$^{-1}$ for N–$P_2O_5$–$K_2O$, respectively. |
| Irrigation water | EC: 1.8 mS cm$^{-1}$ pH 6.9 $H_2PO_4$: <0.63 mg L$^{-1}$ $NO_3^-$: <1 mg L$^{-1}$ $K^+$: 7.19 mg L$^{-1}$ | EC: 1.3 mS cm$^{-1}$ pH 7.4 $H_2PO_4$: 0.89 mg L$^{-1}$ $NO_3^-$: 1.12 mg L$^{-1}$ $K^+$: 6.40 mg L$^{-1}$ |
| Groundwater [26] | Piezometric level close to 1 m. Dry residue between 2000 and 3500 mg L$^{-1}$. EC: > 5.5 mS cm$^{-1}$. | |
| Climate conditions | The climate in the Region of Murcia is dry Mediterranean type and belongs to the Köppen "Bsh" classification, with mild winters and dry and very hot summers. The average annual temperature is close to 22.5 °C, with low rainfall of less than 300 mm and an annual reference evapotranspiration of 1435 mm [22,27]. | |

[1] CEC: cation exchange capacity.

## 2.2. Field Measurement

Plant water status was evaluated as (i) stem water potential at solar midday ($\Psi_S$) using a Scholander-type pressure chamber model Pump-Up (PMS Instrument Company,

Albany, OR, USA) covering the leaves with foil bags 2 h before measurement and as (ii) leaf gas exchange parameters: leaf stomatal conductance (Lc) and net photosynthesis (Pn) at solar midday using a portable gas exchange system CIRAS-2 (PP-Systems, Hitchin, Hertfordshire, UK). The established $CO_2$ concentration was $\approx 400$ µmol mol$^{-1}$, and the photosynthetic photon flux density was 1200 µmol m$^{-2}$ s$^{-1}$. Temperature and relative humidity corresponded to the environment during the measurements. $\Psi_S$, Lc, and Pn were measured in three adults leaves per replicate, the first mature leaf from the apex ($n = 12$ per treatment) every 7–10 days in 2020 and at the end of the 2021 trial.

The evolution of the soil water matric potential was determined using a sensor model Teros-21 (METER Group, Pullman, WA, USA) in three replicates ($n = 3$ per treatment). The sensors were installed at 20 cm depth and 10 cm from the dripper, in the wetting bulb closest to the plant. The minimum observed daily value was used for the analyses. The sensor data were acquired every minute and averaged every 15 min.

The soil volumetric water content was measured every 10 cm, between 10 and 60 cm depth, using an FDR-type probe model Drill & Drop (Sentek Technologies, Stepney, Australia) per replicate ($n = 3$ per treatment). The sensors have an accuracy of $\pm 0.03\%$ and the manufacturer's calibration curve ($R^2 = 0.97$) was used [28]. The probes were installed at 10 cm from the dripper in the wetting bulb closest to the plant. Data obtained were normalized to their field capacity at each depth ($\theta_{FC}$; m$^3$ m$^{-3}$).

To determine the yield (t ha$^{-1}$), eight adjacent and marked plants were harvested per replicate ($n = 32$ per treatment) according to the technical-commercial harvesting criteria. All harvested fruits were weighed, measured, and counted individually. Fruit load was determined as kg of fruit per linear m of crop row. In addition, to evaluate the effect of treatments on fruit shape, the diameters of an ellipse were determined: longitudinal (2a) and equatorial (2b); subsequently, the area was determined as a $\times$ b $\times$ π. Finally, to evaluate fruit sphericity, the quotient between b and a was calculated according to Cohen [29].

An unmanned aerial vehicle model Matrice 600 Pro (DJI Technology Inc., Shenzhen, China) equipped with a multispectral sensor RedEdge-MX$^{TM}$ (MicaSense$^\circledR$, Seattle, WA, USA) flew over Farm 1 on 2020 55 DAT and Farm 2 on 2021 60 DAT. The sensor captures five spectral bands: blue (475 nm in the centre and 20 nm bandwidth), green (560 nm in the centre and 20 nm bandwidth), red (R, 668 nm in the centre and 10 nm bandwidth), red edge (717 nm in the centre and 10 nm bandwidth), and near infrared (NIR, 840 nm in the centre and 40 nm bandwidth). The normalized difference vegetation index (NVDI) [30] was calculated according to Equation (1):

$$\text{NDVI} = (\text{NIR} - \text{R}) / (\text{NIR} + \text{R}) \tag{1}$$

Ground cover was calculated from the drone image, as the percentage of ground area assigned to each plant that has been covered by the crop. The spatial resolution of the images was 2.5 cm $\times$ 2.5 cm per pixel.

*2.3. Fruit Quality Traits*

To assess fruit quality, ten fruits per replicate were randomly selected ($n = 40$ per treatment) at the main harvest. First, flesh firmness as kg cm$^{-2}$ was measured on two sides of each fruit with a hand-held fruit pressure meter model FT011 (TR Scientific Instruments, Forli, Italy) equipped with a 5 mm diameter plunger, and the flesh thickness was measured on two opposite sides of each fruit with a digital caliper (Mitutoyo Co., Kawasaki, Japan). Subsequently, the mesocarp of these fruits was homogenized and filtered to obtain the juice sample for each replicate. The total soluble solids content (TSS; °Brix) was determined using a hand-held refractometer model N-1E (ATAGO, Tokyo, Japan). The juice pH was determined in a 50 mL sample with a digital pH-meter probe in a benchmeter model PCD-6500 (Thermo Fisher Scientific Inc., Sunnyvale, CA, USA), and the titratable acidity (TA) was determined by titrating 10 mL of the extract with 0.1 N NaOH and expressed as g L$^{-1}$ of citric acid. The TSS, pH, and TA were measured immediately in triplicate for each replicate ($n = 12$ per treatment).

The ascorbic acid concentration was measured in a sample of juice stored at $-40\,^{\circ}$C following the protocol described by Kampfenkel et al. [31]. Ascorbic acid was used as standard, and its absorbance was determined at 525 nm. The results were expressed as mg of ascorbic acid per 100 g of fresh weight. All reagents used were analytical quality.

Post-harvest quality was evaluated in a weight loss storage trial, through incubation in a chamber ($4\,^{\circ}$C) followed by room temperature for 15 days each. Four fruits per replicate ($n = 16$) were measured at a 3-day interval. Relative humidity and incubation temperature were controlled using a thermohygrometer Log210 (Dostmann electronic GmbH, Wertheim, Germany).

### 2.4. Irrigation Water Productivity and Nutrient Use Efficiency

Irrigation water productivity ($WP_I$) was determined as kg of fruit per m$^3$ applied [32,33]. The nutrient use efficiency for N, $P_2O_5$, and $K_2O$ was calculated using the partial factor productivity of applied nutrient index (PFP) according to Equation (2) [34]:

$$PFP = Y/F, \tag{2}$$

where Y = yield (kg ha$^{-1}$) and F = amount of nutrient applied (kg ha$^{-1}$).

The nitrate and phosphate content of the irrigation water were both below 1 ppm; therefore, they were not taken into account for the calculations. The potassium input from irrigation water ($K_{Farm\ 1}$: 7.19 mg L$^{-1}$, $K_{Farm\ 2}$: 6.40 mg L$^{-1}$) was considered.

### 2.5. Leaf, Fruit, and Soil Nutrients

Nutritional analysis was carried out in 2020 in leaves, fruit, and soil. Twenty mature and young leaves per replicate ($n = 80$ per treatment) were randomly selected at 46, 60, and 90 DAT. The leaves were washed in a series of 0.1%, Tween$^{®}$ 20 detergent, 1% HCl (0.1 N), and distilled water and then dried at $60\,^{\circ}$C to constant weight, ground, and sieved to a particle diameter of 0.2 mm.

Five fruits ($n = 20$ per treatment) were randomly selected at harvest (90 DAT). The fruits were cut into small pieces and dried at $60\,^{\circ}$C to constant weight and then ground and sieved.

Soil sampling was carried out (i) prior to the trial start, without any irrigation treatments at 30 cm depth, and (ii) at harvest, 90 DAT, at depths of 30 and 60 cm.

Total N was determined on an elemental analyzer model CN628 (LECO Corporation, St. Joseph, MI, USA), and macronutrients (P and K) content was carried out using the method of inductively coupled plasma atomic emission spectroscopy (ICP-AES), based on the standard UNE-EN 15510 [35].

### 2.6. Statistical Analysis

Prior to the ANOVA, assumptions were tested: the normality of the error distributions of each dependent variable was evaluated according to the Shapiro–Wilk test ($p < 0.05$), and the homoscedasticity of the variances was evaluated with the Levene test ($p < 0.05$), using absolute residuals to minimize the possible effect of outliers and improve the power of the test [36–38]. For variables that did not fit a normal distribution, the data were log-transformed. Finally, when significant differences between treatments were detected, means were separated via Duncan's test ($p < 0.05$). All the statistical analysis were carried out using Infostat software version 2020e [39].

## 3. Results

### 3.1. Irrigation Scheduling through Sensors

The weather conditions were very similar between the two years of the trial in the two farms. During the 2020 crop season, the average reference evapotranspiration ($ET_0$) was 5.07 mm day$^{-1}$, while in 2021, it was 4.41 mm day$^{-1}$, ranging between 2.2 and 6.4 mm day$^{-1}$ and between 0.95 and 6.67 mm day$^{-1}$, for the first and second year, respectively. The cumulative rainfall during the experimental period was 46 and 131 mm for the first and

second year, respectively. In 2021, almost half of this rainfall occurred on 23rd May, with almost 60 mm. The VPD values averaged 0.89 and 0.81 kPa, for 2020 and 2021, respectively (Figure 1).

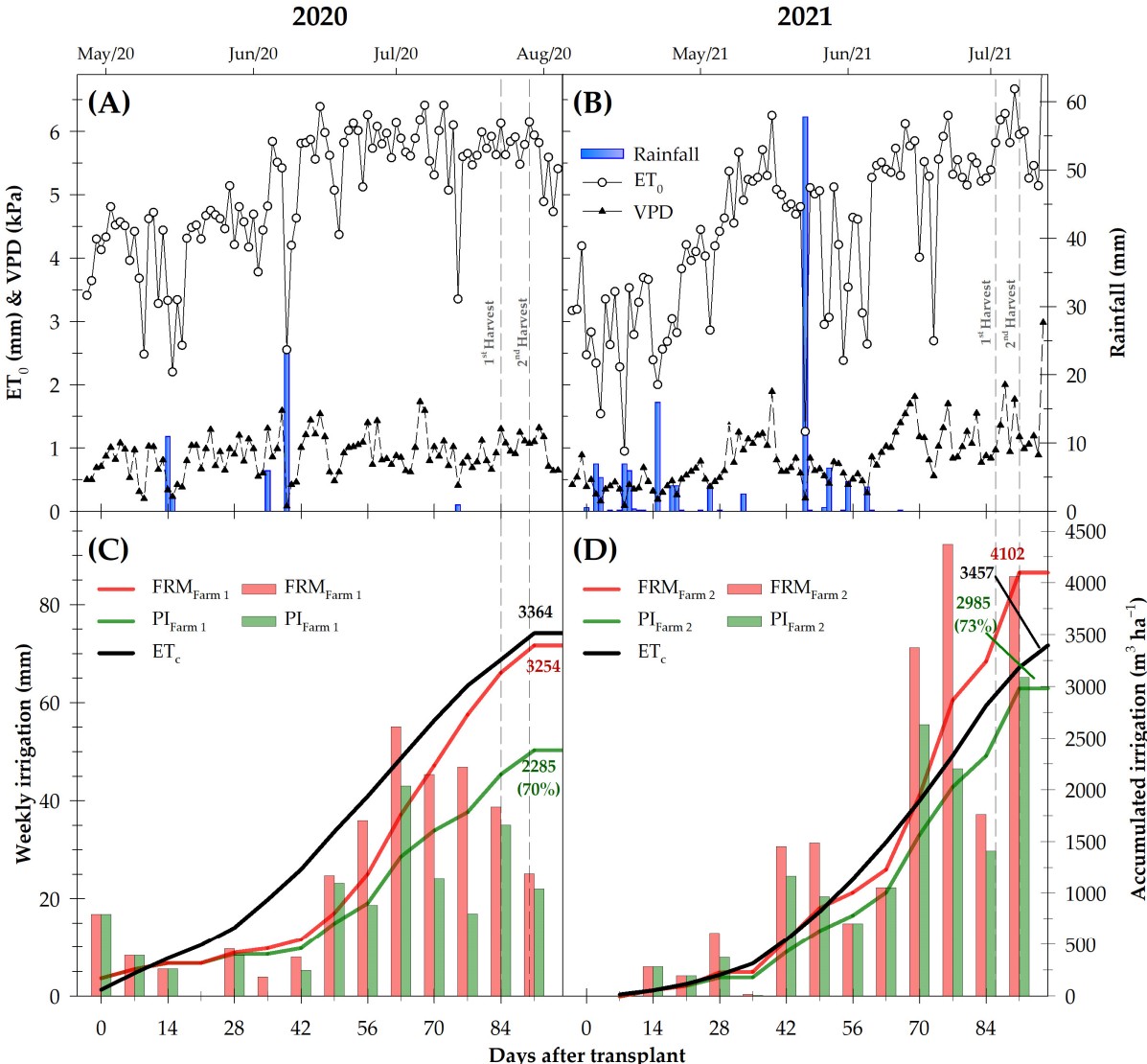

**Figure 1.** Daily evolution of climatic parameters: reference crop evapotranspiration ($ET_0$), vapor pressure deficit (VPD), and rainfall for each year (**A**,**B**). Weekly (bars) and accumulated (lines) irrigation for each treatment (**C**,**D**). Numbers in brackets indicate the irrigation water used percentage with respect to the FRM each year. 0 DAT corresponds to 30 April in 2020 (Farm 1) or 7 April in 2021 (Farm 2).

Figure 1 shows the evolution of the water applied per irrigation treatment in the two farms, with respect to the water that the irrigated crop would receive to meet the water requirements using FAO methodology (100% $ET_c$). Weekly irrigation applied in the two seasons, to both treatments (FRM and PI), was similar until 27 or 25 days after transplant (DAT), respectively, in 2020 and 2021 (during vegetative development), with inputs between 5 and 15 mm per week, coinciding with an evaporative demand between 2.2 and 5.1 mm of $ET_0$ per day in 2020. Water applied in 2021 was slightly lower in this period, due to lower $ET_0$ values, ranging between 0.9 and 4.45 mm day$^{-1}$.

Once $ET_0$ reached 5 mm per day, on 50 DAT in 2020 and 42 DAT in 2021, weekly applied water increased to 30 m$^3$ ha$^{-1}$. In 2020, from 60 DAT onwards, the climatic demand

increased to around 6 mm per day, and so did the applied water, at around 50 m$^3$ ha$^{-1}$, decreasing from day 84 onwards because of the harvest.

Regarding the precision irrigation treatment in 2020, the reduction of water applied related to the farmer irrigation increased as the season progressed, in order to maintain the maximum allowable depletion initially planned. At the end of the season, PI treatment totalled 2285 m$^3$ ha$^{-1}$, being 29.8% less than the FRM treatment (Figure 1). The reduction of applied water coincided with phenological phases from flowering to ripening.

In 2021, the water applied did not follow a continuous rising curve due to rainfall and a lower climatic demand occurred between 49 and 63 DAT. Compared to the previous year, the water corresponding to 100% ET$_c$ was slightly higher (101 mm, +3%); both the farmer and precision treatments applied more water than in 2020, about 848 and 700 m$^3$ ha$^{-1}$, respectively, due to a higher VPD in the last 3 weeks of season (+23.3%) and the higher number of plants per ha in the second year (Table 1), although the coverage rates were similar between both. Thus, the weekly applied water in the farmer treatment ranged between 20 and 90 mm in the second part of the growing season and between 20 and 65 mm for the PI treatment, totalling 4102 and 2985 m$^3$ ha$^{-1}$ for FRM and PI treatment, respectively, (Figure 1) at the end of the crop cycle, obtaining a water saving of 29.8 and 27.2% in the PI treatment for both seasons compared to FRM.

### 3.2. Soil and Plant Water Status

Figure 2 shows the evolution of the minimum soil water matric potential (SWMP) values at 20 cm depth, in which both treatments showed values below –20 kPa, a value corresponding to the relationship of field capacity—maximum soil water retention capacity. The most negative values in 2020, reaching –200 kPa, corresponded to those of the PI treatment, at the maximum permissible depletion of water in the soil, reaching a value of around 25% with respect to the field capacity (Figure 3).

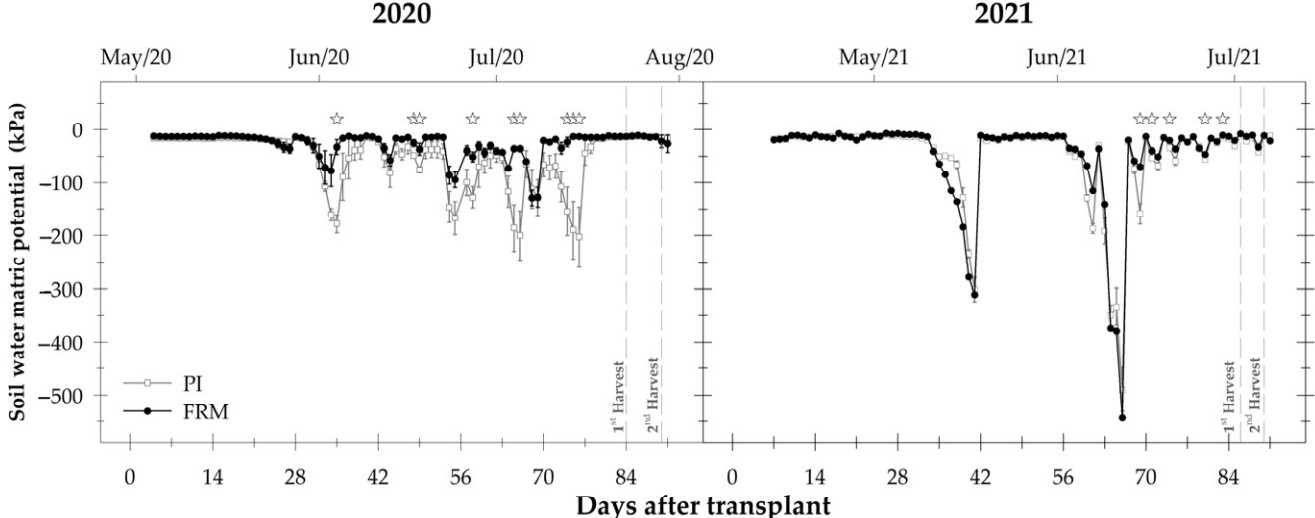

**Figure 2.** Daily evolution of the minimum soil water matric potential at 20 cm depth for 2020 and 2021. Stars indicate significant differences according to the ANOVA ($p < 0.05$). Means ± standard error, $n = 4$.

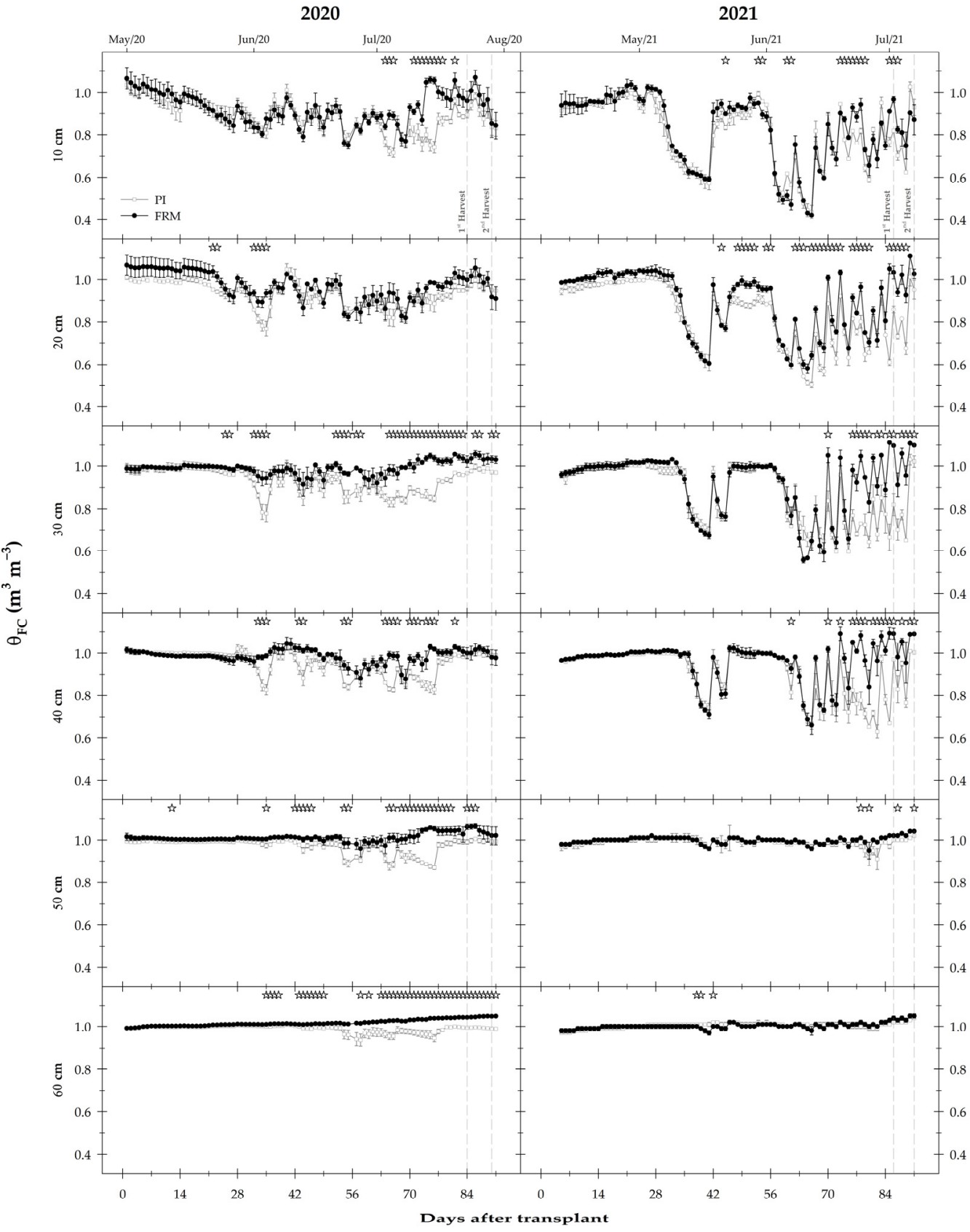

**Figure 3.** Daily evolution of the minimum soil volumetric water content between 10 and 60 cm depth, relative to field capacity for the same depth and day. The stars indicate significant differences according to the ANOVA ($p < 0.05$). Means ± standard error, $n = 4$.

In 2021, the two most important reductions in the SWMP values were due to a breakage in the sprinkler head two times, between 30–40 DAT and 53–60 DAT, in which plants were subjected to a high water demand, up to 6.25 mm per day.

Figure 3 shows the evolution of the minimum volumetric soil water content values in the soil profile with respect to field capacity ($\theta_{FC}$), from 10 to 60 cm depth, in the two years of study.

It can be appreciated that there is a greater soil water depletion in the PI treatment (white line) than in the control (black line). Soil water depletion in PI reached minimum values of around 20% in 2020 and 30% in 2021, starting on 30 DAT in 2020 or 49 DAT in 2021 and increasing in both years at the end of the season, according to the increase in the water reduction applied with respect to FRM.

The irrigation protocol followed in 2020 could not be exactly replicated in 2021 due to both the heavy rainfall on 46 DAT and the break in the irrigation head between 30 and 40 and 53 and 60 DAT. However, no water and nutrients leaching were detected below the root system during the two-year experimental period. The leaching was monitored through the values obtained via sensors; there was no soil water content increase in the deepest soil layers when irrigated (Figure S1).

It should be noted that the volumetric water content in the soil at the greatest depth denoted the high piezometric height of the aquifer. During 2021, both treatments, FRM and PI, were near to field capacity during the whole season; nonetheless, in 2020, we can observe water depletion even at 60 cm depth.

The leaf gas exchange parameters (stomatal conductance and net photosynthesis) were evaluated during 2020 and at the end of 2021 for the two irrigation treatments (Table S1), showing no significant differences between treatments. While net photosynthesis values ranged between 16.0 and 25.5 $\mu$mol m$^{-2}$ s$^{-1}$, leaf conductance values ranged between 107 and 290 mmol m$^2$ s$^{-1}$ for the 2020 season. At the end of 2021, the values for net photosynthesis were around 27 and 28.5 $\mu$mol m$^{-2}$ s$^{-1}$ and for leaf conductance 381 and 440 mmol m$^{-2}$ s$^{-1}$. Although neither of these parameters show significant differences between treatments, in 2020 the leaf conductance trended to show lower values than in 2021, with a similar deficit applied. Stem water potential values were around $-0.5$ MPa for both years, showing no differences between treatments (Table S1).

Different multispectral indices have not shown significant differences in plant vigor (NVDI) between the irrigation treatments during the two crop years; however, they show interannual differences (Table S2). Vegetative growth measured as crop ground cover did not show differences between treatments either (Table S2).

### 3.3. Fertilization Reduction

As irrigation was reduced in PI compared to FRM treatment, fertilization applied through fertigation was reduced too. Figure 4 shows the evolution of the nutrients supplied in each irrigation treatment during the experimental period. The nutrients were supplied through the drip irrigation system, via fertigation, so their application was subject to climatic demand and the occurrence of rainfall, according to the irrigation treatments. Thus, in 2020, for the FRM treatment, the nutrients applied were 60.6, 27.7, and 35.5 kg ha$^{-1}$, for N, $P_2O_5$, and $K_2O$, respectively; in 2021, the inputs were higher, specifically 75.3, 26.6, and 65 kg ha$^{-1}$, for N, $P_2O_5$, and $K_2O$. In 2020, weekly nutrient application ranged between 2.3 and 14 kg ha$^{-1}$ for N, 1.1 and 4.7 kg ha$^{-1}$ for $P_2O_5$, and 1.3 and 8.3 kg ha$^{-1}$ for $K_2O$. In 2021, these increased to maximum values of 23, 7.5, and 15 kg ha$^{-1}$ for N, $P_2O_5$, and $K_2O$ per week, respectively. The PI treatment reduced nutrient inputs according to the reduction of the irrigation water applied. The savings in N–$P_2O_5$–$K_2O$ fertiliser were 39.7%, 37.4%, and 19.1% and 22.7%, 16.2%, and 25.7% during 2020 and 2021, respectively.

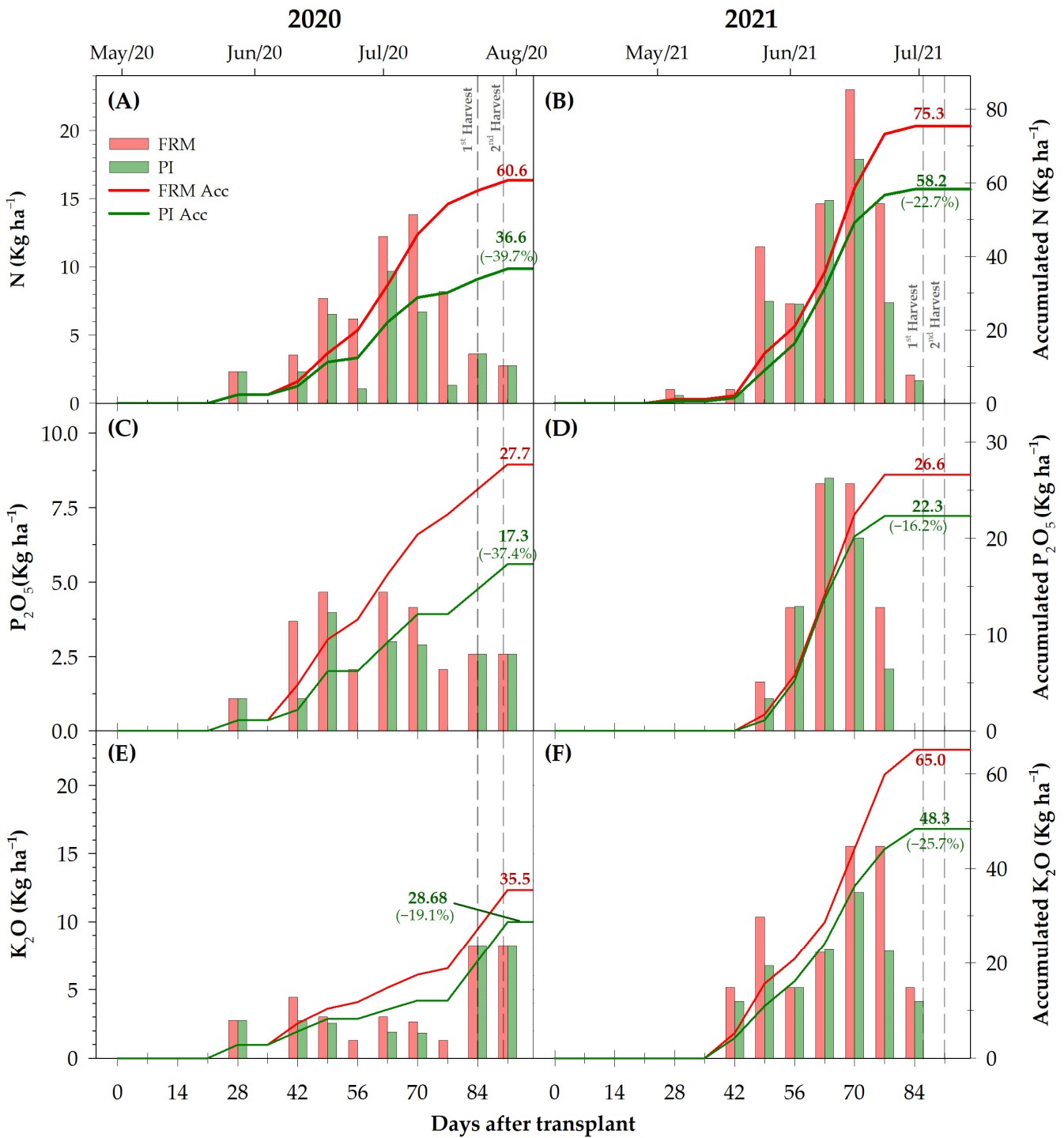

**Figure 4.** Weekly dose of nutrients (N–$P_2O_5$–$K_2O$) applied through fertilisation in melon plants subjected to different irrigation regimes (FRM: farmer criteria and PI: precision irrigation) during 2020 (**A,C,E**) and 2021 (**B,D,F**). The numbers in brackets indicate the percentage savings with respect to the FRM treatment for each year. 0 DAT corresponds to 30 April in 2020 (Farm 1) or 7 April in 2021 (Farm 2).

*3.4. Soil and Plant Nutritional Status*

Table S3 shows the concentration of the main macronutrients in the leaf, during the growing season on 46, 60, and 90 DAT. Foliar N and K extraction decreased with crop growth, showing significant differences between irrigation treatments, with leaf N

extraction being 9% lower in PI than in FRM at 60 DAT. Leaf P concentration was significantly affected by the irrigation treatment from day 60 onward, ranging from 0.20 to 0.51%. The rest of the foliar nutrients analysed showed no significant differences between irrigation treatments.

At harvest (90 DAT), the concentration of different macro- and micro-nutrients were not affected by irrigation treatment (Table S4), neither were the different forms of nitrogen analysed (Table S4).

The soil nutritional analysis can be found in Table S5. Although fertigation was affected in the different irrigation treatments (Section 3.3), the nutritional status of the soil was not affected in those treatments (Table S5). Referring to the initial analysis, a high consumption of soil total nitrogen (NT) was observed during crop growth in both treatments. $P_2O_5$ and $K_2O$ consumption was slightly higher in PI treatment at root depth. Regarding the rest of the soil parameters analysed, it is worth highlighting the decrease in organic matter and increase in cation exchange capacity (CEC) during the growing season, with no difference between irrigation treatments.

### 3.5. Harvest and Fruit Quality

Table 2 shows the results for yield, number of fruits per m, and individual fruit weight. Two harvests were carried out commercially in each trial, at 84 and 90 DAT in 2020 or 86 and 91 DAT in 2021. At the seasonal level, the yields achieved in 2021 were significantly higher than in the previous season, almost doubling the total harvested, due to the significant increase in fruit weight and the higher plant density, although the fruit load measured as fruit per plant did not differ between years (Table 2). No differences were detected in the precocity or total yield of the crop due to the effect of reduced water and nutritional intake in both years of study. The total fruit load and medium fruit size did not vary between treatments, with the exception of the fruit load during the first harvest of 2021.

**Table 2.** Production parameters obtained in melon plants subjected to different irrigation regimes (FRM: farmer criteria and PI: precision irrigation) during 2020 and 2021.

| Year | DAT | Treatment | Yield (t ha$^{-1}$) | Fruit Load (Fruits per Plant) | Fruit Weight (kg per Fruit) |
|------|-----|-----------|---------------------|-------------------------------|------------------------------|
| 2020 | 84 | FRM | 5.89 ± 1.71 a | 0.50 ± 0.14 a | 3.39 ± 0.02 a |
| | | PI | 5.55 ± 1.94 a | 0.48 ± 0.17 a | 3.43 ± 0.17 a |
| | 90 | FRM | 15.16 ± 2.29 a | 1.69 ± 0.23 a | 2.57 ± 0.05 a |
| | | PI | 17.07 ± 1.39 a | 2.06 ± 0.21 a | 2.42 ± 0.12 a |
| | Total | FRM | 21.05 ± 2.97 a | 2.23 ± 0.31 a | 2.84 ± 0.47 a |
| | | PI | 22.62 ± 2.46 a | 2.53 ± 0.24 a | 2.59 ± 0.28 a |
| 2021 | 86 | FRM | 29.36 ± 2.50 a | 1.80 ± 0.12 a | 3.77 ± 0.10 a |
| | | PI | 37.47 ± 4.20 a | 2.55 ± 0.27 b | 3.44 ± 0.09 a |
| | 91 | FRM | 12.64 ± 4.88 a | 1.05 ± 0.39 a | 2.78 ± 0.11 a |
| | | PI | 4.82 ± 0.71 a | 0.48 ± 0.08 a | 2.59 ± 0.10 a |
| | Total | FRM | 41.99 ± 7.04 a | 2.88 ± 0.46 a | 3.43 ± 0.04 a |
| | | PI | 42.28 ± 4.20 a | 3.00 ± 0.27 a | 3.31 ± 0.11 a |
| | | Year (Y) | *** | ns | *** |
| | | Treatment (T) | ns | ns | ns |
| | | Y × T | ns | ns | ns |

DAT: Days after transplanting. Means ± standard error, $n = 4$. Different letters for the same parameter, DAT, and year indicate significant differences according to Duncan's test ($p < 0.05$). Asterisks indicate differences in total harvest for year (Y), treatment (T), and Y × T. ***: $p < 0.001$ and ns: non-significant, for the ANOVA.

Table 3 shows the main fruit quality characteristics at the main harvest in each of the two seasons, for the two irrigation treatments. The different parameters evaluated showed similar values between the two irrigation treatments, except for the total soluble solids (TSS) values, which were slightly higher in the FRM treatment, significantly in 2021, and ascorbic acid, which were higher in the PI irrigation, 24 and 37% for each year, respectively.

**Table 3.** Fruit quality traits obtained in melon plants subjected to different irrigation regimes (FRM: farmer criteria and PI: precision irrigation) during 2020 and 2021.

| Year | Treatment | TSS (°Brix) | TA (g L$^{-1}$) | pH - | Firmness (kg cm$^{-2}$) | Ascorbic Acid (mg 100 g$^{-1}$ FW) |
|---|---|---|---|---|---|---|
| 2020 | FRM | 13.5 ± 0.6 a | 0.65 ± 0.04 a | 6.11 ± 0.05 a | 5.18 ± 0.13 a | 7.97 ± 0.46 b |
| | PI | 13.0 ± 0.2 a | 0.69 ± 0.03 a | 6.08 ± 0.04 a | 5.07 ± 0.15 a | 9.93 ± 0.63 a |
| 2021 | FRM | 13.3 ± 0.1 a | 0.89 ± 0.03 a | 6.09 ± 0.01 b | 5.07 ± 0.13 a | 6.15 ± 0.48 b |
| | PI | 12.5 ± 0.1 b | 0.72 ± 0.09 a | 6.22 ± 0.03 a | 5.05 ± 0.17 a | 8.85 ± 0.96 a |
| Year (Y) | | ns | * | ns | ns | * |
| Treatment (T) | | ns | ns | ns | ns | ** |
| Y × T | | ns | ns | * | ns | ns |

TSS: total soluble solids, TA: titratable acidity, and FW: fresh weight. Means ± standard error, $n = 4$. Different letters for the same parameter and year indicate significant differences according to Duncan's test. Asterisks indicate differences for year (Y), treatment (T), and Y × T ($p < 0.05$). **: $p < 0.01$, *: $p < 0.05$ and ns: non-significant, for the ANOVA.

Titratable acidity (TA) did not show significant differences between treatments but did differ according to the year of study. The maturity index (relationship TSS/TA) was not different between treatments, following the same pattern as the TA. Melon juice pH was affected by the irrigation treatment in 2021, increasing in PI to 6.22, while slightly decreasing the value of its organic acids as measured via titratable acidity. Pulp firmness was not affected by the treatment, not even under the 2021 higher fruit load conditions.

The ascorbic acid (vitamin C) concentration maintains a trend inversely proportional to the irrigation applied. The vitamin C gets higher as the irrigation is reduced (PI vs. FRM) and is significantly lower in 2021 than 2020, on average, where 770 m$^3$ ha$^{-1}$ more were applied.

During the experimental period, the PI treatment showed higher irrigation water (WP$_I$) and nutrients (PFP) use efficiency values than the FRM treatment. It is worth noting that the PI treatment increased the WP$_I$ values by 38 and 53% for each year with respect to the farmer treatment. In the case of nitrogen and phosphate use efficiency (Table 4), the values increased by 78 and 72%, respectively, in 2020 and 30 and 20% in 2021, although only significant differences were found in 2020. Potassium use efficiency, although not significant within years, shows a pronounced trend to increase in the PI treatment with respect to the FRM treatment (Table 4).

**Table 4.** Irrigation water productivity (WP$_I$) and applied nutrients partial factor productivity index (PFP), obtained in melon plants subjected to different irrigation regimes (FRM: farmer criteria and PI: precision irrigation) during 2020 and 2021.

| Year | Treatment | WP$_I$ (kg m$^{-3}$) | PFP (t kg N $^{-1}$) | (t kg P$_2$O$_5$ $^{-1}$) | (t kg K$_2$O $^{-1}$) |
|---|---|---|---|---|---|
| 2020 | FRM | 6.47 ± 0.91 b | 0.347 ± 0.049 b | 0.761 ± 0.107 b | 0.358 ± 0.050 a |
| | PI | 9.90 ± 1.08 a | 0.619 ± 0.067 a | 1.307 ± 0.142 a | 0.502 ± 0.054 a |
| 2021 | FRM | 10.24 ± 1.71 b | 0.558 ± 0.094 a | 1.578 ± 0.263 a | 0.476 ± 0.077 a |
| | PI | 14.16 ± 1.41 a | 0.728 ± 0.073 a | 1.895 ± 0.188 a | 0.628 ± 0.062 a |
| Year (Y) | | * | ns | ** | ns |
| Treatment (T) | | * | * | * | * |
| Y × T | | ns | ns | ns | ns |

Means ± standard error, $n = 4$. Different letters for the same parameter and year indicate significant differences according to Duncan' test ($p < 0.05$). Asterisks indicate differences for year (Y), treatment (T), and Y × T. **: $p < 0.01$, *: $p < 0.05$ and ns: non-significant, for the ANOVA.

The water loss measured in the post-harvest experiment, showed no significant changes after cold storage or room temperature storage in either of the storage temperatures tested (25 °C in 2020 or 17 °C in 2021), although the fruits of the FRM treatment showed higher values than those of the PI treatment, with average values around 6.5 and

5.2%, for 2020 and 2021, respectively, while the PI treatment values were lower, 5.2 and 4.9%, respectively.

## 4. Discussion

Irrigation and nutrient scheduling (fertigation) based on the use of sensors that provide real-time soil water status information allowed water depletion to be maintained in the soil in the melon root growth area. This soil water depletion was partially stable during the experimental period, reaching minimum values of around 20% and 30% with respect to field capacity in 2020 and 2021, respectively.

Yield was not negatively affected, with no differences in fruit load (fruit per m) or fruit weight (kg) between irrigation treatments, although in the second year a higher yield was obtained due to seasonal changes.

The accumulated water savings in each year relative to the farmer (FRM) treatment ranged between 30 and 27% for 2020 and 2021, respectively, reaching values of 2285 and 2985 $m^3$ $ha^{-1}$ in the precision irrigation (PI) treatment (Figure 1). The FRM-applied water differed annually mainly due to the 2021 higher VPD values (Figure 1) and a higher number of plants per hectare (Table 1). Despite this, the crop cover was similar in both years (Table S2).

The water requirements quantified for the open field growing melon, depending on the climate of the area, varied between 4000 and 4500 $m^3$ $ha^{-1}$ [40,41]. The difference in annual water requirements in this study highlights the increase in water productivity using irrigation scheduling based on soil water control, as compared to that based on crop coefficient estimation [20,42,43].

Likewise, crop water status indicators (stem water potential, net photosynthesis, and stomatal conductance) allowed it to be determined that the irrigation reduction applied did not negatively affect crop water status, since no differences were found between both treatments. Due to the high variability presented, stem water potential and leaf conductance values were not comparable with other melon trials within the same variety, as could be observed in Chevilly et al. [16]. However, other studies have shown that after prolonged stress over time leaf water potential could reach values as low as –2 MPa [44]. Regarding leaf conductance, Ribas et al. [41] concluded that Lc is not a good indicator of water stress for this crop after daily irrigation, although it is used by other authors to justify the stress applied [44]. Regarding the carbon assimilation values obtained, our plants showed values comparable to those obtained in other melon trials under adequate water supply [44,45].

The irrigation scheduling carried out reduced water and nutrient inputs throughout the whole soil profile, minimizing any leaching below the root system depth and reducing the damaging effects of a high piezometric level of the aquifer on the root system (Figures 1 and 3). In fact, the irrigation reduction applied in the PI treatment also implied a similar magnitude reduction of fertilization, as shown in the values observed in the evolution of the nutrients supplied (Figure 4). Specifically, savings in N–$P_2O_5$–$K_2O$ fertiliser were 39.7%, 37.4%, and 19.1% and 22.7%, 16.2%, and 25.7%, for 2020 and 2021, respectively.

Nutritional inputs to the crop in the farm treatment have been lower than those considered optimal in other studies conducted in this variety by Castellanos et al. [46,47], where 93 to 155 kg N $ha^{-1}$ were applied.

All foliar macronutrients, besides nitrogen, remained within the expected values for this crop at the different growth stages [45,48,49], without having been affected by the reduction in fertilization contribution.

The N foliar extraction decreased along with crop growth, when it begins to translocate into the fruit, as shown by other authors [46,50]. Although all the values were in the optimum range reported for melon leaves [51,52], punctually, in the leaf to fruit translocation period, it showed significant differences between irrigation treatments, foliar N in PI being 9% lower than with FRM irrigation.

At harvest, neither the nutritional quality of the fruit nor the soil reserves were affected between treatments. But soil nutrients have been reduced compared to the initial analysis,

with a considerable decrease in total nitrogen and organic matter during the crop cycle, the crop ending with values of 0.05 and 0.6%, respectively. It should be noted that the initial values for nitrogen in the soil are considered low [53], and the problem of soil organic matter levels in the area has been highlighted, being "very low" from the initial values of the crop [12]. It should be mentioned that the plant material remains (Table S3) are returned into the soil once the crop is finished, so that both parameters in the soil can be reestablished for the next growing season.

Regarding harvest quality, there was no difference in titratable acidity (TA), total soluble solids (TSS), firmness, flesh thickness, or nutritional parameters analysed in both treatments. Melon juice pH was affected by the irrigation treatment in 2021, but it always remained within the range for 'Piel de Sapo' melon [54]. Therefore, we can conclude that the product obtained has been of similar quality in the two irrigation strategies established.

Ascorbic acid concentration presented in the fruit juice were similar to those found in other reports for 'Piel de Sapo' melon [55], in this type of melon being much higher than that found in others [45,54,56]. Vitamin C concentration was affected by irrigation treatment, obtaining a higher concentration in the PI treatment with respect to the FRM treatment in both years. This concentration was proportional to the reduction of irrigation applied, decreasing in the second year with respect to the first, with higher irrigation applied (Table 3 ad Figure 1).

Ascorbic acid synthesis has recently been shown to be promoted under drought conditions in other crops [57], being one of the major antioxidant molecules involved in plant cell protection from oxidative stress [58,59]. In the present study, in spite of the absence of significant differences between irrigation treatments in the plant water status indicators analysed, the soil water status, measured as both soil water matric potential and soil volumetric water content, has been found to be significantly different during most of the season. Thus, we can underline the high sensitivity of organic acids as early indicators of water stress.

## 5. Conclusions

Monitoring the soil water depletion at values of around 20% to 30% through real-time sensors allowed the irrigation water inputs to be reduced by an average of 30%, increasing water productivity up to 3.92 kg m$^{-3}$ more than farmer irrigation. Even the functional quality of the fruit was notably improved, as the concentration of organic acids, such as vitamin C increased. Likewise, since water and nutrients were supplied jointly through fertigation, nutrient application was reduced by a similar percentage to that of irrigation water. This trial demonstrated that through the monitoring of soil water content, the sustainability of crops growing under semi-arid conditions can be improved.

**Supplementary Materials:** The following supporting information can be downloaded at: https://www.mdpi.com/article/10.3390/agronomy13102652/s1, Figure S1. Continuous evolution of volumetric soil water content in the soil profile from 10 to 60 cm depth, for the two irrigation treatments, at an interval of 13 and 9 days during summer, for the years 2020 and 2021, respectively. Table S1. Evolution of gas exchange parameters in the trial period for melon plants subjected to different irrigation regimes (FRM: farmer criteria and PI: precision irrigation): Net photosynthesis (Pn), leaf conductance (Lc), and midday stem water potential ($\Psi_S$); Table S2. Multispectral indexes and crop ground cover for melon plant subjected to different irrigation regimes (FRM: farmer criteria and PI: precision irrigation); Table S3. Evolution of foliar NPK in melon plants subjected to different irrigation regimes (FRM: farmer criteria and PI: precision irrigation) in 2020; Table S4. Harvested melon macro- and micro-nutrients, ammoniacal nitrogen, and nitrate in plants subjected to different irrigation regimes (FRM: farmer criteria and PI: precision irrigation) in 2020; Table S5. Bulk soil analysis for melon plants subjected to different irrigation regimes (FRM: farmer criteria and PI: precision irrigation) in 2020.

**Author Contributions:** Conceptualization, A.P.-P.; methodology, A.P.-P., P.J.E., C.M., and S.Z.-G.; validation, A.T., S.Z.-G., and P.B.; formal analysis, A.T., P.B., and S.Z.-G.; investigation, S.Z.-G., A.T., A.P.-P., and P.B.; resources, A.T., S.Z.-G., C.M., and P.J.E.; data curation, S.Z.-G., A.T., P.B., and A.P.-P.; writing—original draft preparation, S.Z.-G. and A.P.-P.; writing—review and editing, S.Z.-G., A.P.-P., and P.J.E.; visualization, S.Z.-G. and P.B; supervision, A.P.-P., C.M., and S.Z.-G.; project administration, A.P.-P. and C.M.; funding acquisition, A.P.-P. All authors have read and agreed to the published version of the manuscript.

**Funding:** This work is a result of the AGROALNEXT programme and was supported by MCIN with funding from European Union NextGenerationEU (PRTR-C17.I1) and by Fundación Séneca with funding from Comunidad Autónoma Región de Murcia (CARM). Funding has also been received from the FMC Agricultural Sciences chair of the UPCT, an agreement between FMC Agricultural Solutions and UPCT, and with the Sindicato Central de Regantes del Acueducto Tajo-Segura.

**Data Availability Statement:** Data will be made available on request.

**Acknowledgments:** The authors thank to the Irrigation Community of Campo de Cartagena for letting them use the facilities to carry out the study. Susana Zapata-García acknowledges her research fellowship of the FMC Agriculture Sciences chair with Universidad Politécnica de Cartagena.

**Conflicts of Interest:** The authors declare no conflict of interest.

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
