# Peer review of "Using Soil Water Status Sensors to Optimize Water and Nutrient Use in Melon under Semi-Arid Conditions"

_agronomy, doi:10.3390/agronomy13102652_

Round 1
Reviewer 1 Report
The paper presents the efficiency of water and nutrient use in irrigation scheduling using multi-deep soil sensors in melon crops in semi-arid conditions. Two crop cycles were carried out with full and deficient irrigation. Soil, climate, and crop data were monitored daily throughout the cycle.
The manuscript is very well written, in direct and understandable English. The experimental methodology is clearly detailed allowing the experiment to be replicated. The models and manufacturers of the instruments used were described.
The measurement units are adequate, although some are not in the form of scientific notation, which was highlighted in yellow in the file attached to the system.
Only two equations are cited and presented in the text, but the numbering is wrong, which has already been duly highlighted in the attached file.
All tables and figures are properly cited in the text, as well all citations are listed in the references section.
The results are well presented and represent a good potential for scientific contribution.
The manuscript is suitable for publication after minor revisions and adjustments which are pointed out in the manuscript revised file (see attached at the system).

Author Response
REVIEWER 1 COMMENTS
The paper presents the efficiency of water and nutrient use in irrigation scheduling using multi-deep soil sensors in melon crops in semi-arid conditions. Two crop cycles were carried out with full and deficient irrigation. Soil, climate, and crop data were monitored daily throughout the cycle.
The manuscript is very well written, in direct and understandable English. The experimental methodology is clearly detailed allowing the experiment to be replicated. The models and manufacturers of the instruments used were described.
The measurement units are adequate, although some are not in the form of scientific notation, which was highlighted in yellow in the file attached to the system.
Only two equations are cited and presented in the text, but the numbering is wrong, which has already been duly highlighted in the attached file.
All tables and figures are properly cited in the text, as well all citations are listed in the references section.
The results are well presented and represent a good potential for scientific contribution.
The manuscript is suitable for publication after minor revisions and adjustments which are pointed out in the manuscript revised file (see attached at the system).
Thank you for your review and suggestions.
Comments in the attached file:
- Lines 79-80. Insert the acronym: (VPD)
Done.
Lines [82-87]
The crop evapotranspiration (ETc) was calculated according to the FAO method [20] and the crop coefficients (Kc) for the initial, mid-season and end stages, were 0.5, 0.85 and 0.6, respectively [21]. Reference evapotranspiration (ET0), rainfall and vapor pressure deficit (VPD) were obtained from the agroclimatic station ‘Torre Pacheco TP-42’ belonging to the “Murcia Agrometeorological Information Service” network (37°46’26” N 0°53’55” W) [22].
- Table 1. add the unit
- Table 1. use scientific notation in all units of measurement.
- Line 162. Use Four instead of 4.
Done. Thank you for the corrections.
Line [109]
Table 1. Experimental conditions of each commercial farm used during the study years.
|
|
Farm 1 |
Farm 2 |
|
|
|
|
|
Year |
2020 |
2021 |
|
|
|
|
|
Location |
Torre Pacheco (Region of Murcia, SE Spain) |
|
|
|
37°45'58" N 0°58'03" W |
37°47'18" N 1°2'54" W |
|
Cultivar |
Cordial F1 (Sakata Seeds) |
Valderas F1 (Clause Vegetable Seeds) |
|
|
|
|
|
Growing cycle |
90 days (30 April to 29 July) |
91 days (07 April to 07 July) |
|
|
|
|
|
Cultivation system |
Geotextile micro tunnel of polypropylene fibers for thermal protection (0.5 m high in the middle) and transparent plastic mulch. |
|
|
1.8 m × 1.6 m planting frame 3472 plants per ha |
1.8 m × 1.3 m planting frame 4273 plants per ha |
|
|
|
|
|
|
Irrigation system |
Drip irrigation system with one drip line per row and emitters spaced at 0.3 m with a flow rate of 2.2 L h−1. |
|
|
Standard cultural practices |
The fertilization, weed, pest and disease control program were carried out according to commercial management by the usual criteria for the productive zone. |
|
|
|
|
|
|
Soil characteristics |
The soil profile up to 0.3 m depth corresponded to a clay loam texture class, with a bulk density of 1.40 g cm-3 and 1.4 % of organic matter. 1CEC was 11.2 meq/100g and the soil nutrients concentrations were 1.07 – 0.11 – 0.59 g Kg−1 for N – P2O5 – K2O, respectively. |
The soil profile up to 0.3 m depth corresponded to a clay loam texture class, with a bulk density of 1.35 g cm-3 and 1.7 % of organic matter. 1CEC was 16.5 meq/100g and the soil nutrients concentrations were 1.16 – 0.09 – 0.33 g Kg−1 for N – P2O5 – K2O, respectively. |
|
|
|
|
|
Irrigation water |
EC: 1.8 mS cm−1 pH 6.9 H2PO4: <0.63 mg L−1 K+: 7.19 mg L−1 |
EC: 1.3 mS cm−1 pH 7.4 H2PO4: 0.89 mg L−1 K+: 6.40 mg L−1 |
|
|
|
|
|
Groundwater [25] |
Piezometric level close to 1 m Dry residue between 2000 and 3500 mg L−1. EC: > 5.5 mS cm−1 |
|
|
|
|
|
|
Climate conditions |
The climate in the Region of Murcia is dry Mediterranean type and belongs to the Köppen “Bsh” classification, with mild winters and dry and very hot summers. The average annual temperature is close to 22.5 °C, low rainfall of less than 300 mm and an annual reference evapotranspiration of 1435 mm [22,26]. |
|
1CEC: cation exchange capacity.
- Line 140. Eq. 2 is Eq. 1
Corrected. The number of equations, tables and figures was revised.
Lines [148-152]
Normalized Difference Vegetation Index (NVDI) [28] was calculated according to Eq. 1:
|
|
(1) |
Ground cover was calculated from the drone image, as the percentage of ground area assigned to each plant that has been covered by the crop. The spatial resolution of the images was 2.5 cm x 2.5 cm per pixel.
- Line 157. Do not use a comma to separate subject from verb.
Done.
Lines [167-170]
The ascorbic acid concentration was measured in a sample of juice stored at −40 °C following the protocol described by Kampfenkel et al [29]. Ascorbic acid was used as standard, and its absorbance was determined at 525 nm. The results were expressed as mg of ascorbic acid per 100 g of fresh weight. All reagents used were analytical quality.
- Line 162. (ºC) Replace the ordinal number symbology with degree symbology.
- Line 387. (ºC) Replace the ordinal number symbology with degree symbology.
- Table 3. (ºBrix) Replace the ordinal number symbology with degree symbology.
Thanks for the correction, the symbology has been checked throughout the manuscript again.
Lines [171-174]
Post-harvest quality was evaluated in a weight loss storage trial, through incubation in chamber (4 °C) followed by room temperature for 15 days each. Four fruits per replicate (n = 16) were measured in a 3-day interval. Relative humidity and incubation temperature were controlled by a thermohygrometer Log210 (Dostmann electronic GmbH, Germany).
Lines [399-404]
The water loss measured in the post-harvest experiment, showed no significant changes after cold storage or room temperature storage in neither of both storage temperatures tested (25 °C in 2020 or 17 °C in 2021), although the fruits of the FRM treatment showed higher values than those of the PI treatment, with average values around 6.5 and 5.2%, for 2020 and 2021, respectively, while PI were lower, 5.2 and 4.9% respectively (data not shown).
Line [373]
Table 3. Fruit quality traits obtained in melon plants subjected to different irrigation regimes (FRM: farmer criteria and PI: precision irrigation) during 2020 and 2021.
|
Year |
Treatment |
TSS |
TA |
pH |
Firmness |
Ascorbic acid |
|
(°Brix) |
(g L−1) |
- |
(Kg cm−2) |
(mg 100 g−1 FW) |
||
|
|
|
|
|
|
|
|
|
2020 |
FRM |
13.5 ±0.6 a |
0.65 ±0.04 a |
6.11 ±0.05 a |
5.18 ±0.13 a |
7.97 ±0.46 b |
|
|
PI |
13.0 ±0.2 a |
0.69 ±0.03 a |
6.08 ±0.04 a |
5.07 ±0.15 a |
9.93 ±0.63 a |
|
|
|
|
|
|
|
|
|
2021 |
FRM |
13.3 ±0.1 a |
0.89 ±0.03 a |
6.09 ±0.01 b |
5.07 ±0.13 a |
6.15 ±0.48 b |
|
|
PI |
12.5 ±0.1 b |
0.72 ±0.09 a |
6.22 ±0.03 a |
5.05 ±0.17 a |
8.85 ±0.96 a |
|
|
|
|
|
|
|
|
|
|
|
|
|
|
|
|
|
Year (Y) |
ns |
* |
ns |
ns |
* |
|
|
Treatment (T) |
ns |
ns |
ns |
ns |
** |
|
|
Y ⨯ T |
ns |
ns |
* |
ns |
ns |
|
|
|
|
|
|
|
|
|
TSS: total soluble solids, TA: titratable acidity and FW: fresh weight. Means ± standard error, n = 4. Different letters for the same parameter and year indicate significant differences according to Duncan’s test. Asterisks indicates differences for year (Y), treatment (T) and Y × T (p < 0.05). **: p < 0.01, *: p < 0.05 and ns: non-significant, for the ANOVA.
- Line 169. Eq. 3 is Eq. 2
Done.
Lines [177-180]
Irrigation water productivity (WPI) was determined as Kg of fruit per m3 applied [30,31]. The nutrient use efficiency for N, P2O5 and K2O, was calculated using the partial factor productivity of applied nutrient index (PFP) according to Eq. 2 [32]:
|
, |
(2) |
where, Y = yield (Kg ha–1) and F = amount of nutrient applied (Kg ha–1).
- Line 255 & Line 275. Means ± standard error. Not shown in the figure.
Thanks for your comment, they are shown, although this measure have shown little variability withing the treatment in some periods. We have made adjustments to the figure to improve the display of data.
- Line 346: **: p < 0.01, *: p < 0.05. This caption is unnecessary as it is not included in the table.
Done.
Lines [357-360]
DAT: Days after transplanting. Means ± standard error, n = 4. Different letters for the same parameter, DAT and year indicate significant differences according to Duncan’s test (p < 0.05).
Asterisks indicates differences in total harvest for year (Y), treatment (T) and Y × T. ***: p < 0.001 and ns: non-significant, for the ANOVA.
- Table 3. ‘0 DAT correspond to 30th April 362 in 2020 or 07th April in 2021’ & ‘***: p < 0.001’. This caption is unnecessary as it is not included in the table.
Done.
Lines [375-377]
TSS: total soluble solids, TA: titratable acidity and FW: fresh weight. Means ± standard error, n = 4. Different letters for the same parameter and year indicate significant differences according to Duncan’s test.
Asterisks indicates differences for year (Y), treatment (T) and Y × T (p < 0.05). **: p < 0.01, *: p < 0.05 and ns: non-significant, for the ANOVA.
- Lines 373-375. Nitrogen and Phosphate use efficiency. It is necessary to make it clear that there was no significant difference in 2021, as described for K.
Lines [386-390]
Done, also the paragraph was rephrased for its better understood.
In the case of nitrogen and phosphate use efficiency (Table 4), the values get increased by 78 and 72% respectively in 2020 and 30 and 20% in 2021, although only significant differences were found in 2020. Potassium use efficiency although not significant within years, shows a pronounced trend to increase in PI treatment with respect to FRM (Table 4).
- Table 4. WPi. I expected there to be a significant difference between treatments in each year....Are you sure they are the same letters?
- Table 4. ***: p < 0.001. This caption is unnecessary as it is not included in the table.
Thank you very much for the correction, indeed, differences between treatments were detected as indicated in the effect of (T).. Also, unnecessary captions were removed.
Lines [391]
Table 4. Irrigation water productivity (WPI) and applied nutrients partial factor productivity index (PFP), obtained in melon plants subjected to different irrigation regimes (FRM: farmer criteria and PI: precision irrigation) during 2020 and 2021.
|
Year |
Treatment |
WPI |
PFP |
||
|
(Kg m−3) |
(t Kg N −1) |
(t Kg P2O5 −1) |
(t Kg K2O −1) |
||
|
|
|
|
|
|
|
|
2020 |
FRM |
6.47 ±0.91 b |
0.347 ±0.049 b |
0.761 ±0.107 b |
0.358 ±0.050 a |
|
|
PI |
9.90 ±1.08 a |
0.619 ±0.067 a |
1.307 ±0.142 a |
0.502 ±0.054 a |
|
|
|
|
|
|
|
|
2021 |
FRM |
10.24 ±1.71 b |
0.558 ±0.094 a |
1.578 ±0.263 a |
0.476 ±0.077 a |
|
|
PI |
14.16 ±1.41 a |
0.728 ±0.073 a |
1.895 ±0.188 a |
0.628 ±0.062 a |
|
|
|
|
|
|
|
|
|
|
|
|
|
|
|
Year (Y) |
* |
ns |
** |
ns |
|
|
Treatment (T) |
* |
* |
* |
* |
|
|
Y ⨯ T |
ns |
ns |
ns |
ns |
|
|
|
|
|
|
|
|
Means ± standard error, n = 4. Different letters for the same parameter and year indicate significant differences according to Duncan’ test (p < 0.05).
Asterisks indicates differences for year (Y), treatment (T) and Y × T. **: p < 0.01, *: p < 0.05 and ns: non-significant, for the ANOVA.
- Line 425. Figure. Add "s" (plural).
Done.
Lines [437-440]
The irrigation scheduling carried out reduced water and nutrient inputs throughout the whole soil profile, minimizing any leaching below the root system depth and reducing the damaging effects of a high piezometric level of the aquifer on the root system (Figures 1 & 3).

Reviewer 2 Report
The paper has a sound structure and is clearly written. Also, it can be read easily. However, I have missed in "Introduction" some references regarding the state of the art in water/nutrient use in melon and what the paper address comparing what it is already known.
Many references are cited in "Discussion" and it would be worthy to cite them first in "Introduction".
Other comments are shown within the file attached.

The English style is good but there are still some long paragraphs that needs to be rewriten to improve clarity.
Author Response
REVIEWER 2 COMMENTS
The paper has a sound structure and is clearly written. Also, it can be read easily. However, I have missed in "Introduction" some references regarding the state of the art in water/nutrient use in melon and what the paper address comparing what it is already known.
Many references are cited in "Discussion" and it would be worthy to cite them first in "Introduction".
Other comments are shown within the file attached.
Comments in the attached file:
Abstract
- Line 17. 2 two
- Line 26. Please specify what does "not affected negatively" mean.
- Line 26. , on average,
- Line 27. change "," by ";"
Done. Thank you for your suggestions.
Lines [13-28]
Abstract: Nowadays, agriculture must satisfy the growing demand for food, and increasing its sustainability, from an environmental, economic and social point of view, is the only way to achieve this. The objective of this study was to increase the water and nutrient use efficiency of a melon crop during two consecutive seasons under commercial conditions, growing under semi-arid area. For this purpose, two treatments were studied: i) a farmer treatment (FRM), fertigated at ~100% of crop evapotranspiration (ETc) during the whole growing season; and ii) a precision irrigation treatment (PI), irrigated by adjusting, between flowering and ripening, the weekly farmer irrigation to minimize the leaching below the root system. The threshold for allowable soil water depletion in the active root uptake zone was set at 20–30 %. The cumulative water savings in each year relative to the FRM treatment ranged between 30 and 27% for 2020 and 2021, respectively. Yield was not negatively affected, with no differences in fruit load (fruit per m) or fruit weight (Kg) between irrigation treatments, although higher yields were obtained in the second year due to seasonal changes. The crop water status indicators evaluated (stem water potential, net photosynthesis and stomatal conductance) were not affected by the irrigation treatments. Water and nitrogen productivity, on average, increased by 45.5 and 54.4% during the experimental period respectively; the average PI ascorbic acid content increased 33.4%.
Introduction
- Line 33. in general - delete.
- Line 39. plants'
- Please, cite any report already published on the effect of water/nutrient use efficiency.
Thanks for the suggestions, we have complemented the introduction with references to other related research and suitable amendments.
Lines [60-65]
Water is a vital resource for human life and ecosystems biodiversity, but also indispensable for sustainability of economic activities and development [1]. However, there is a high pressure on the use of water resources which will be exacerbated by the climate change. The irrigated agricultural sector is the most demanding of water, accounting for about a 70% of global water withdrawals [2] and is expected to face a complex challenge. Due to the rising temperatures, semi-arid Mediterranean areas such as the southeast of Spain will be the most affected, as they will have to extend the irrigation period of the crops, at the same time that plants’ evapotranspiration will increase [3–6].
Along with water scarcity, we must face an increase in world population, with a growing demand for food. That fact has come to the attention of the United Nations, creating different Sustainable Development Goals [7] related to agriculture and rational water use. This underscores the need of recognizing the value of water in agriculture and make a rational and sustainable use of water, through different measures to optimize its use and management, leading to a sustainable intensification of the agriculture [8,9].
The melon crop (Cucumis melo L.) would be particularly sensitive to drought and water stress, as it would negatively affect leaf growth and cause fruit cracking, reducing the yield [10] assuming therefore large economic losses. Spain is the main producer and exporter in the European Union, generating 63% of the European production, that amount totals a value of 325 M EUR [11].
The Region of Murcia, in spite of being located in a semi-arid area [12], is the main national producer for plain skin melons, a category that includes ‘Piel de Sapo’ and Yellow melons, which account for 68% of Spanish production in 2021 [13]. These areas, which have had a steady diminishing supply of water, have been able to maintain a strong agricultural activity, that not only supply the food necessary to feed a growing population, also play an important role mitigating climate change through the carbon sequestration in the soil [14]. These facts evidence the need to increase the agriculture sustainability to ensure the sector’s activity, and improving the water use efficiency will minimize the environmental impact and increase the productivity of agricultural systems to achieve economic, social, and environmental benefits. Regarding this need, some authors as Fabeiro et al. have analyzed the sensitivity of the different phenological stages of melon to moderate or severe water deficit through controlled deficit irrigation techniques [15], or by identifying drought resistant traits to improve breeding strategies [16]. In addition to water, the nutrient use efficiency in melon has been studied through fertilization at different doses, as shown in Castellanos et al [17].
Materials and Methods
- Table 1. . The average annual ....
- Line 269. Please specified which is the depth of the water table.
Done. Also, we have included groundwater data in table 1.
Lines [109]
Table 1. Experimental conditions of each commercial farm used during the study years.
|
|
Farm 1 |
Farm 2 |
|
|
|
|
|
Year |
2020 |
2021 |
|
|
|
|
|
Location |
Torre Pacheco (Region of Murcia, SE Spain) |
|
|
|
37°45'58" N 0°58'03" W |
37°47'18" N 1°2'54" W |
|
Cultivar |
Cordial F1 (Sakata Seeds) |
Valderas F1 (Clause Vegetable Seeds) |
|
|
|
|
|
Growing cycle |
90 days (30 April to 29 July) |
91 days (07 April to 07 July) |
|
|
|
|
|
Cultivation system |
Geotextile micro tunnel of polypropylene fibers for thermal protection (0.5 m high in the middle) and transparent plastic mulch. |
|
|
1.8 m × 1.6 m planting frame 3472 plants per ha |
1.8 m × 1.3 m planting frame 4273 plants per ha |
|
|
|
|
|
|
Irrigation system |
Drip irrigation system with one drip line per row and emitters spaced at 0.3 m with a flow rate of 2.2 L h−1. |
|
|
Standard cultural practices |
The fertilization, weed, pest and disease control program were carried out according to commercial management by the usual criteria for the productive zone. |
|
|
|
|
|
|
Soil characteristics |
The soil profile up to 0.3 m depth corresponded to a clay loam texture class, with a bulk density of 1.40 g cm-3 and 1.4 % of organic matter. 1CEC was 11.2 meq/100g and the soil nutrients concentrations were 1.07 – 0.11 – 0.59 g Kg−1 for N – P2O5 – K2O, respectively. |
The soil profile up to 0.3 m depth corresponded to a clay loam texture class, with a bulk density of 1.35 g cm-3 and 1.7 % of organic matter. 1CEC was 16.5 meq/100g and the soil nutrients concentrations were 1.16 – 0.09 – 0.33 g Kg−1 for N – P2O5 – K2O, respectively. |
|
|
|
|
|
Irrigation water |
EC: 1.8 mS cm−1 pH 6.9 H2PO4: <0.63 mg L−1 K+: 7.19 mg L−1 |
EC: 1.3 mS cm−1 pH 7.4 H2PO4: 0.89 mg L−1 K+: 6.40 mg L−1 |
|
|
|
|
|
Groundwater [25] |
Piezometric level close to 1 m. Dry residue between 2000 and 3500 mg L−1. EC: > 5.5 mS cm−1. |
|
|
|
|
|
|
Climate conditions |
The climate in the Region of Murcia is dry Mediterranean type and belongs to the Köppen “Bsh” classification, with mild winters and dry and very hot summers. The average annual temperature is close to 22.5 °C, low rainfall of less than 300 mm and an annual reference evapotranspiration of 1435 mm [22,26]. |
|
1CEC: cation exchange capacity.
- Line 121. Cite the probe's precision. Do they were calibrated "in-situ" for the local soil conditions?
Under the experimental conditions of a commercial farm, we do not consider a local calibration necessary, for four main reasons: (i) the soil is constantly being disturbed for organic matter incorporation and tillage between crop cycles, (ii) the horticultural crop cycles are too short, (iii) the irrigation criterion is established based on water dynamics in the soil, and finally, (iv) the default calibration curve used has a high fit (R2 0.97) for clay loam texture class our experimental conditions. Also, to add repeatability to our results we relativised the values to the soil field capacity.
Nevertheless, we have indicated and referenced the calibration curve used and the accuracy of the sensor.
Lines [128-133]
The soil volumetric water content was measured every 10 cm, between 10 and 60 cm depth, using an FDR-type probe model Drill & Drop (Sentek Technologies, Australia) per replicate (n = 3 per treatment). The sensors have an accuracy of ±0.03% and the manufacturer's calibration curve (R2 = 0.97) for the clay loam texture class was used [28]. The probes were installed at 10 cm from the dripper in the wetting bulb closest to the plant. Data obtained were normalized to their field capacity at each depth (qFC; m3 m-3).
- Line 139. What was the spatial resolution of the images?
Done.
Lines [150-152]
Ground cover was calculated from the drone image, as the percentage of ground area assigned to each plant that has been covered by the crop. The spatial resolution of the images was 2.5 cm x 2.5 cm per pixel.
- Line 206. volume was due to the – delete
Done.
Lines [216-218]
In 2021, almost half of this rainfall occurred on 23rd May, with almost 60 mm. The VPD values averaged 0.89 and 0.81 kPa, for 2020 and 2021, respectively (Figure 1).
- Line 271. Do you have checked what was the water movement within the soil after irrigation? Does the water front reached depths beyond 50 cm in all the irrigations?
It will be interesting to show these results. They could explain why the SWC in the deepest horizons almost did not change and where more and less alike.
As supplementary material (Figure 1S), the evolution of the soil volumetric water content at different depths for the two treatments was added, showing that the PI treatment did not detect the water applied with irrigation from a depth of 50 cm. The following text has been added to explain how irrigation time has been controlled, to avoid promoting leaching.
Lines [218-280].
The leaching was monitored thorough the values obtained by sensors, regarding there were no soil water content increase in the deepest soil layers when irrigated (Figure S1).
Figure S1. Continuous evolution of volumetric soil water content in the soil profile from 10 to 60 cm depth, for the two irrigation treatments, at an interval of 13 and 9 days during summer, for the years 2020 and 2021, respectively.
Discussion.
- Lines 393 & 412. us – delete
- Line 420. In terms of– Regarding
Done.
Lines [434-436]
Irrigation and nutrient scheduling (fertigation) based on the use of sensors that provide real-time soil water status information, has allowed to maintain water depletion in the soil in the melon root growth area. This soil water depletion has been partially stable during the experimental period, reaching minimum values of around 20% and 30% with respect to field capacity in 2020 and 2021, respectively.
Likewise, crop water status indicators (stem water potential, net photosynthesis, and stomatal conductance) have allowed to determine that the irrigation reduction applied did not negatively affect crop water status, since no differences were found between both treatments. Due to the high variability presented, stem water potential and leaf conductance values were not comparable with other melon trials withing the same variety, as could be observed in Chevilly et al. [16]. Although other studies have shown that after prolonged stress over time leaf water potential could reach values as low as –2 MPa [42]. Regarding to leaf conductance, Ribas et al. [39] concluded that Lc is not a good indicator of water stress for this crop after daily irrigation, although is used for other authors to justify the stress applied [42]. Regarding the carbon assimilation values obtained, our plants showed values comparable to those obtained in other melon trials under adequate water supply [42,43].
- Line 416 Chevilly reference - I do not think this author is cited in the introduction.
The introduction section has been improved, including this author’s reference amongst others.
- Line 450. In terms of– Regarding
Done.
Lines [464-469]
Regarding harvest quality, there was no difference in titratable acidity (TA), total soluble solids (TSS), firmness, flesh thickness or nutritional parameters analysed in both treatments. Melon juice pH was affected by the irrigation treatment in 2021, but it always remained within the range for ‘Piel de Sapo’ melon [52]. Therefore, we can conclude that the product obtained has been of similar quality in the two irrigation strategies established.
Conclusions.
- Line 473. us – delete
- Line 475. in terms of an increase in the – delete
- Line 477. very – delete
- Line 478. change , to .
- Line 479. Rewrite the last sentence. It is not clear.
Lines [486-493]
Monitoring the soil water depletion at values of around 20% to 30% through real time sensors has allowed to reduce the irrigation water inputs by an average of 30%, increasing water productivity up to 3.92 Kg m−3 more than farmer irrigation. Even the functional quality of the fruit has been notably improved, as the concentration of organic acids, such as vitamin C have increased. Likewise, since water and nutrients have been supplied jointly through fertigation, nutrient application has been reduced in a similar percentage to that of irrigation water. This trial remarked that though the monitoring of soil water content, the sustainability of crops growing under semi-arid conditions can be improved.

Reviewer 3 Report
1、Supplement farmland groundwater level data of the field in table 1 .
2、The precision irrigation treatment (PI) was not accurately described。 Such as depth of active root zone 。did 20-30% present the percent of field capacity or available soil moisture?
3、water moisture usually decrease with depth under drip irrigation , why it increased in Fig 3 at depth of 50 and 60cm below? The higher moisture would lead to leaching of water and nutrients which contradicts the following conclusion s。Pls check the data in the fig.
4、Check the accuracy of the error analysis results in Table 2 or if they were correctly marked. such as 2.55 ± 0.27b and 1.80 ±0.12 , or 1.05 ±0.39 a and 0.48 ±0.08 a in the same column.
5、the study was useful but the trail design with only two treatments was was too simple.
no advice
Author Response
REVIEWER 3 COMMENTS
- 1、Supplement farmland groundwater level data of the field in table 1.
Done. Included a field in table 1 with groundwater data from a technical report of the Geological and Mining Institute of Spain (IGME).
Lines [109]
Table 1. Experimental conditions of each commercial farm used during the study years.
|
|
Farm 1 |
Farm 2 |
|
|
|
|
|
Year |
2020 |
2021 |
|
|
|
|
|
Location |
Torre Pacheco (Region of Murcia, SE Spain) |
|
|
|
37°45'58" N 0°58'03" W |
37°47'18" N 1°2'54" W |
|
Cultivar |
Cordial F1 (Sakata Seeds) |
Valderas F1 (Clause Vegetable Seeds) |
|
|
|
|
|
Growing cycle |
90 days (30 April to 29 July) |
91 days (07 April to 07 July) |
|
|
|
|
|
Cultivation system |
Geotextile micro tunnel of polypropylene fibers for thermal protection (0.5 m high in the middle) and transparent plastic mulch. |
|
|
1.8 m × 1.6 m planting frame 3472 plants per ha |
1.8 m × 1.3 m planting frame 4273 plants per ha |
|
|
|
|
|
|
Irrigation system |
Drip irrigation system with one drip line per row and emitters spaced at 0.3 m with a flow rate of 2.2 L h−1. |
|
|
Standard cultural practices |
The fertilization, weed, pest and disease control program were carried out according to commercial management by the usual criteria for the productive zone. |
|
|
|
|
|
|
Soil characteristics |
The soil profile up to 0.3 m depth corresponded to a clay loam texture class, with a bulk density of 1.40 g cm-3 and 1.4 % of organic matter. 1CEC was 11.2 meq/100g and the soil nutrients concentrations were 1.07 – 0.11 – 0.59 g Kg−1 for N – P2O5 – K2O, respectively. |
The soil profile up to 0.3 m depth corresponded to a clay loam texture class, with a bulk density of 1.35 g cm-3 and 1.7 % of organic matter. 1CEC was 16.5 meq/100g and the soil nutrients concentrations were 1.16 – 0.09 – 0.33 g Kg−1 for N – P2O5 – K2O, respectively. |
|
|
|
|
|
Irrigation water |
EC: 1.8 mS cm−1 pH 6.9 H2PO4: <0.63 mg L−1 K+: 7.19 mg L−1 |
EC: 1.3 mS cm−1 pH 7.4 H2PO4: 0.89 mg L−1 K+: 6.40 mg L−1 |
|
|
|
|
|
Groundwater [25] |
Piezometric level close to 1 m. Dry residue between 2000 and 3500 mg L−1. EC: > 5.5 mS cm−1. |
|
|
|
|
|
|
Climate conditions |
The climate in the Region of Murcia is dry Mediterranean type and belongs to the Köppen “Bsh” classification, with mild winters and dry and very hot summers. The average annual temperature is close to 22.5 °C, low rainfall of less than 300 mm and an annual reference evapotranspiration of 1435 mm [22,26]. |
|
1CEC: cation exchange capacity.
- 2、The precision irrigation treatment (PI) was not accurately described。 Such as depth of active root zone 。did 20-30% present the percent of field capacity or available soil moisture?
Thank you for your comment. We have improved the description of the treatment.
Lines [96-108]
A randomized experimental design was established for both farms with four repetitions, each one composed by three adjacent rows of eight plants. The central row was monitored as the experimental unit and the others served as plants border. Two irrigation treatments with four replicates each, were tested: (i) farmer (FRM), irrigated to satisfying crop needs during the entire cycle, according to the ETc; and (ii) a precision irrigation treatment (PI), irrigated by adjusting, between flowering and ripening, the weekly farmer irrigation to minimize the leaching below the root system. The threshold for the permissible soil water depletion in the zone of active root absorption (30 cm depth) was set between 20-30% of field capacity. The evolution of the soil water content (SWC) was measured with FDR-type sensors. As the fertigation was applied during the irrigation events, the reduction in fertilizers dose was proportional to the reduction in irrigation water for each day. The amount of water applied in each treatment was controlled by volumetric water meters.
- 3、water moisture usually decrease with depth under drip irrigation , why it increased in Fig 3 at depth of 50 and 60cm below? The higher moisture would lead to leaching of water and nutrients which contradicts the following conclusion s。Pls check the data in the fig.
As supplementary material (Figure 1S), the evolution of the soil volumetric water content at different depths for the two treatments was added, showing that the PI treatment did not detect the water applied with irrigation from a depth of 50 cm.
Although the reduction of the wetting front with depth is clear, the moisture values at depth are higher than at surface, because the piezometric height of the aquifer is very high. This fact has already been specified in table 1.
- 4、Check the accuracy of the error analysis results in Table 2 or if they were correctly marked. such as 55 ± 0.27b and 1.80 ±0.12 , or 1.05 ±0.39 a and 0.48 ±0.08 a in the same column.
The analysis is correct. The letters indicate significant differences for the same parameter, DAT and year; therefore, the comparison is between the two irrigation treatments (FRM & PI) for each condition. Nevertheless, caption has been modified to clarify the data considered for each statistical analysis.
Lines [357-360]
DAT: Days after transplanting. Means ± standard error, n = 4. Different letters for the same parameter, DAT and year indicate significant differences according to Duncan’s test (p < 0.05).
Asterisks indicates differences in total harvest for year (Y), treatment (T) and Y × T. ***: p < 0.001 and ns: non-significant, for the ANOVA.
- 5、the study was useful but the trail design with only two treatments was too simple.
Evaluating further treatments on a commercial farm is more complex per se, which we assume when conducting applied research. We also decided to perform it on two different farms to obtain more accurate conclusions. We do not agree that the number of treatments influences the complexity of an experiment, in contrast to the hypotheses to be tested and the study conditions, which ensure the replicability of our results. Furthermore, the evaluations carried out and the equipment used allow us to strengthen our results. Finally, under these conditions, we have chosen to obtain more replicates than treatments in order to minimise error.
- English very difficult to understand/incomprehensible
We have carefully reviewed English with native speakers. The other three reviewers have indicated that the written English is OK and easy to understand. Nevertheless, we have checked any grammatical errors or wrong definitions again.

Reviewer 4 Report
The manuscript agronomy-2612125 entitled ‘Using soil water status sensors to optimize water and nutrient use in melon under semi-arid conditions’ was conducted to investigate the effect of irrigation scheduling using multi-depth soil sensors on water and nutrient use efficiency of melon crops. Irrigation scheduling have been performed using soil moisture sensor that enables to save water up to 27 – 30% in the two seasons.
It presents good quality experiment with a comprehensive measurements, so I think this manuscript appropriate for agronomy journal.
However, I have some recommendations and observations, as the following:
- Table 1: The common electrical conductivity abbreviation is EC not CE.
- Please present the soil hydro-physics data and soil EC.
- It is not clear in the manuscript how determined the amount of water applied in each irrigation event in the PI treatment to ensure that the leaching not increase.
- Two different cultivars have been used in this work as mentioned in table 1, which may affect the results of the two seasons?
- Line 216-220: it is better to move this sentence to material and method section.
- Table 2 included harvest dates as a factor, although this was not taken into account in the statistical analysis, which causes confusion for the reader. I suggest that only cumulative data be shown.
- The data presented in table 2-4 should be revised in the view of the letters followed the treatment means that must be corresponded to the significance indices (ns or * or **) for Y x T interaction presented in the same table. In this regard, Table2: Fruit load means has different letters, however, its ns as presented below in the same table. Table 3: Please revise pH data (FRM=6.09 has b letter, however, treatment effect not significant). Table 3: the TSS results presented in the text as it significant but the significance indices in Table 3 is ns. Similarly in table 4: N and P and K that not consistent with text (particularly Line 376).
- Line 334-335: harvests dates should be also identified in the material and method section.
No
Author Response
REVIEWER 4 COMMENTS
The manuscript agronomy-2612125 entitled ‘Using soil water status sensors to optimize water and nutrient use in melon under semi-arid conditions’ was conducted to investigate the effect of irrigation scheduling using multi-depth soil sensors on water and nutrient use efficiency of melon crops. Irrigation scheduling have been performed using soil moisture sensor that enables to save water up to 27 – 30% in the two seasons.
It presents good quality experiment with a comprehensive measurements, so I think this manuscript appropriate for agronomy journal.
Thank you for your comments, we appreciate them.
However, I have some recommendations and observations, as the following:
- Table 1: The common electrical conductivity abbreviation is EC not CE.
- Please present the soil hydro-physics data and soil EC.
Done, thanks for the correction. In table 1, we have added information on the soil's hydro-physical properties.
Lines [109]
Table 1. Experimental conditions of each commercial farm used during the study years.
|
|
Farm 1 |
Farm 2 |
|
|
|
|
|
Year |
2020 |
2021 |
|
|
|
|
|
Location |
Torre Pacheco (Region of Murcia, SE Spain) |
|
|
|
37°45'58" N 0°58'03" W |
37°47'18" N 1°2'54" W |
|
Cultivar |
Cordial F1 (Sakata Seeds) |
Valderas F1 (Clause Vegetable Seeds) |
|
|
|
|
|
Growing cycle |
90 days (30 April to 29 July) |
91 days (07 April to 07 July) |
|
|
|
|
|
Cultivation system |
Geotextile micro tunnel of polypropylene fibers for thermal protection (0.5 m high in the middle) and transparent plastic mulch. |
|
|
1.8 m × 1.6 m planting frame 3472 plants per ha |
1.8 m × 1.3 m planting frame 4273 plants per ha |
|
|
|
|
|
|
Irrigation system |
Drip irrigation system with one drip line per row and emitters spaced at 0.3 m with a flow rate of 2.2 L h−1. |
|
|
Standard cultural practices |
The fertilization, weed, pest and disease control program were carried out according to commercial management by the usual criteria for the productive zone. |
|
|
|
|
|
|
Soil characteristics |
The soil profile up to 0.3 m depth corresponded to a clay loam texture class (39% sand, 22% silt, and 39% clay), a bulk density of 1.40 g cm-3, a 1.4% of organic matter and, an EC1:5 of 0.613 mS cm−1. The estimated field capacity and wilting point values based on Saxton et al. [] was 36.1% and 23.8%, respectively. 1CEC was 11.2 meq 100g−1 and the soil nutrients concentrations were 1.07 – 0.11 – 0.59 g Kg−1 for N – P2O5 – K2O, respectively. |
The soil profile up to 0.3 m depth corresponded to a silty clay texture class (14% sand, 44% silt, and 42% clay), a bulk density of 1.35 g cm-3, a 1.7% of organic matter and, an EC1:5 of 0.303 mS cm−1. The estimated field capacity and wilting point values based on Saxton et al. [] was 39.7% and 25.2%, respectively. 1CEC was 16.5 meq 100g−1 and the soil nutrients concentrations were 1.16 – 0.09 – 0.33 g Kg−1 for N – P2O5 – K2O, respectively. |
|
|
|
|
|
Irrigation water |
EC: 1.8 mS cm−1 pH 6.9 H2PO4: <0.63 mg L−1 K+: 7.19 mg L−1 |
EC: 1.3 mS cm−1 pH 7.4 H2PO4: 0.89 mg L−1 K+: 6.40 mg L−1 |
|
|
|
|
|
Groundwater [25] |
Piezometric level close to 1 m. Dry residue between 2000 and 3500 mg L−1. EC: > 5.5 mS cm−1. |
|
|
|
|
|
|
Climate conditions |
The climate in the Region of Murcia is dry Mediterranean type and belongs to the Köppen “Bsh” classification, with mild winters and dry and very hot summers. The average annual temperature is close to 22.5 °C, low rainfall of less than 300 mm and an annual reference evapotranspiration of 1435 mm [22,26]. |
|
1CEC: cation exchange capacity.
- It is not clear in the manuscript how determined the amount of water applied in each irrigation event in the PI treatment to ensure that the leaching not increase.
The following text has been added to explain how irrigation time has been controlled, to avoid promoting leaching.
Lines [278-280]
The leaching was monitored thorough the values obtained by sensors, regarding there were no SWC increase in the deepest soil layers when irrigated.
- Two different cultivars have been used in this work as mentioned in table 1, which may affect the results of the two seasons?
We consider that the use of different cultivars provides additional robustness to the work, as the same methodology is carried out in two different conditions, framed withing the same semiarid environment. It may do affect to the different yield obtained in both seasons; it should be noted that each variety was established at a different planting frame.
- Line 216-220: it is better to move this sentence to material and method section.
The paragraph referred “Weekly irrigation applied in the two seasons, to both treatments (FRM and PI) was similar until 27 or 25 days after transplant (DAT), respectively in 2020 and 2021(during vegetative development), with inputs between 5 and 15 mm per week, coinciding with an evaporative demand between 2.2 and 5.1 mm of ET0 per day in 2020. Water applied in 2021, was slightly lower in this period, due to lower ET0 values, ranging between 0.9 and 4.45 mm day–1.” is a description of the irrigation program results.
In material and method section it is described that “farmer (FRM), irrigated to satisfying crop needs during the entire cycle, according to the ETc; and (ii) a precision irrigation treatment (PI), irrigated by adjusting, between flowering and ripening, the weekly farmer irrigation”, lines 103 to 106.
- Table 2 included harvest dates as a factor, although this was not taken into account in the statistical analysis, which causes confusion for the reader. I suggest that only cumulative data be shown.
The split data of two harvest for each year provide information about the precocity (or not, as it is in this case) of the crop due to the irrigation treatment. Regarding that other trials improve this parameter; we consider that individual harvest data should be included.
To avoid the confusion, now it is indicated in the table 2 foot that just the “Total harvest” data is taken into account for the annual comparative.
- The data presented in table 2-4 should be revised in the view of the letters followed the treatment means that must be corresponded to the significance indices (ns or * or **) for Y x T interaction presented in the same table.
The letters indicate significant differences for the same parameter, DAT and year; therefore, the comparison is between the two irrigation treatments (FRM & PI) for each condition. The factor analysis uses the total harvest data.
Table foot has been modified to clarify this analysis.
In this regard, Table2: Fruit load means has different letters, however, its ns as presented below in the same table.
Average fruit load has no differences between treatments, but it is marked punctually in the first harvest of 2021. As is described in the line 371: “The total fruit load and medium fruit size did not vary between treatments, with the exception of the fruit load during the first harvest of 2021.”
Table 3: Please revise pH data (FRM=6.09 has b letter, however, treatment effect not significant).
We cannot conclude that it is a treatment effect as each year a different tendence is shown (FRM 2020: 6.11a PI 2020:6.08a; FRM 2021: 6.09 b, PI 2021: 6.22 a), therefore, the treatment as a factor (T) has not influence in pH.
Table 3: the TSS results presented in the text as it significant but the significance indices in Table 3 is ns.
The text indicates that those differences are only significant in 2021: “The different parameters evaluated showed similar values between the two irrigation treatments, except for the total soluble solids (TSS) values, which were slightly higher in FRM treatment, significantly in 2021.” – Lines 381 to 383
Similarly in table 4: N and P and K that not consistent with text (particularly Line 376).
To clarify the differences a phrase has been added in line 409: respectively for the nitrogen use efficiency and 72 and 20% for the phosphorus, although only significant differences were found in 2020.
- Line 334-335: harvests dates should be also identified in the material and method section.
These dates have been specified in the results section.
L 334-335. Two harvests were carried out commercially in each trial, at 84 & 90 DAT in 2020 or 86 & 91 DAT in 2021.

Round 2
Reviewer 3 Report
the results was simlpe and should be impeoved ,such as listing your inovations
Reviewer 4 Report
The authors clarified the queries and made most of the suggestions. There are no additional suggestions.